# Speech encoding by coupled cortical theta and gamma oscillations

**Alexandre Hyafil[1]\*, Lorenzo Fontolan[1,2], Claire Kabdebon[1], Boris Gutkin[1,3], Anne-Lise Giraud[2]**

[1]INSERM U960, Group for Neural Theory, Département d'Etudes Cognitives, Ecole Normale Supérieure, Paris, France; [2]Department of Neuroscience, University of Geneva, Geneva, Switzerland; [3]Centre for Cognition and Decision Making, National Research University Higher School, Moscow, Russia

**Abstract** Many environmental stimuli present a quasi-rhythmic structure at different timescales that the brain needs to decompose and integrate. Cortical oscillations have been proposed as instruments of sensory *de-multiplexing*, i.e., the parallel processing of different frequency streams in sensory signals. Yet their causal role in such a process has never been demonstrated. Here, we used a neural microcircuit model to address whether coupled theta–gamma oscillations, as observed in human auditory cortex, could underpin the multiscale sensory analysis of speech. We show that, in continuous speech, theta oscillations can flexibly track the syllabic rhythm and temporally organize the phoneme-level response of gamma neurons into a code that enables syllable identification. The tracking of slow speech fluctuations by theta oscillations, and its coupling to gamma-spiking activity both appeared as critical features for accurate speech encoding. These results demonstrate that cortical oscillations can be a key instrument of speech de-multiplexing, parsing, and encoding.

\*For correspondence: alexandre.
hyafil@gmail.com

**Competing interests:** The authors declare that no competing interests exist.

**Reviewing editor**: Hiram Brownell, Boston College, United States

## Introduction

The physical complexity of biological and environmental signals poses a fundamental problem to the sensory systems. Sensory signals are often made of different rhythmic streams organized at multiple timescales, which require to be processed in parallel and recombined to achieve unified perception. Speech constitutes an example of such a physical complexity, in which different rhythms index linguistic representations of different granularities, from phoneme to syllables and words (*Rosen, 1992*; *Zion Golumbic et al., 2012*). Before meaning can be extracted from continuous speech, two critical pre-processing steps need to be carried out: a de-multiplexing step, i.e., the parallel analysis of each constitutive rhythm, and a parsing step, i.e., the discretization of the acoustic signal into linguistically relevant chunks that can be individually processed (*Stevens, 2002*; *Poeppel, 2003*; *Ghitza, 2011*). While parsing is presumably modulated in a top-down way, by knowing a priori through developmental learning (*Ngon et al., 2013*) where linguistic boundaries should lie, it is likely largely guided by speech acoustic dynamics. It has recently been proposed that speech de-multiplexing and parsing could both be handled in a bottom-up way by the combined action of auditory cortical oscillations in distinct frequency ranges, enabling parallel computations at syllabic and phonemic timescales (*Ghitza, 2011*; *Giraud and Poeppel, 2012*). Intrinsic coupling across cortical oscillations of distinct frequencies, as observed in electrophysiological recordings of auditory cortex (*Lakatos et al., 2005*; *Fontolan et al., 2014*), could enable the hierarchical combination of syllabic- and phonemic-scale computations, subsequently restoring the natural arrangement of phonemes within syllables (*Giraud and Poeppel, 2012*).

The most pronounced energy fluctuations in speech occur at about 4 Hz (*Zion Golumbic et al., 2012*) and can serve as an acoustic guide for signalling the syllabic rhythm (*Mermelstein, 1975*).

**eLife digest** Some people speak twice as fast as others, while people with different accents pronounce the same words in different ways. However, despite these differences between speakers, humans can usually follow spoken language with remarkable ease.

The different elements of speech have different frequencies: the typical frequency for syllables, for example, is about four syllables per second in speech. Phonemes, which are the smallest elements of speech, appear at a higher frequency. However, these elements are all transmitted at the same time, so the brain needs to be able to process them simultaneously.

The auditory cortex, the part of the brain that processes sound, produces various 'waves' of electrical activity, and these waves also have a characteristic frequency (which is the number of bursts of neural activity per second). One type of brain wave, called the theta rhythm, has a frequency of three to eight bursts per second, which is similar to the typical frequency of syllables in speech, and the frequency of another brain wave, the gamma rhythm, is similar to the frequency of phonemes. It has been suggested that these two brain waves may have a central role in our ability to follow speech, but to date there has been no direct evidence to support this theory.

Hyafil et al. have now used computer models of neural oscillations to explore this theory. Their simulations show that, as predicted, the theta rhythm tracks the syllables in spoken language, while the gamma rhythm encodes the specific features of each phoneme. Moreover, the two rhythms work together to establish the sequence of phonemes that makes up each syllable. These findings will support the development of improved speech recognition technologies.

Since the syllabic rate coincides with the auditory cortex theta rhythm (3–8 Hz), syllable boundaries could be viably signalled by a given *phase* in the theta cycle. The relevance of speech tracking by the theta neural rhythm (*Henry et al., 2014*) is highlighted by experimental data showing that speech intelligibility depends on the degree of phase-locking of the theta-range neural activity in auditory cortex (*Ahissar et al., 2001*; *Luo and Poeppel, 2007*; *Peelle et al., 2013*; *Gross et al., 2013*). By analogy with the spatial and mnemonic oscillatory processes that take place in the hippocampus (*Jensen and Lisman, 1996*; *Lisman and Jensen, 2013*; *Lever et al., 2014*), the theta oscillation may orchestrate gamma neural activity to facilitate its subsequent decoding (*Canolty et al., 2007*): the phase of theta-paced neural activity could regulate faster neural activity in the low-gamma range (>30 Hz) involved in linguistic coding of phonemic details (*Ghitza, 2011*; *Giraud and Poeppel, 2012*). The control of gamma by theta oscillations could hence both modulate the excitability of gamma neurons to devote more processing power to the informative parts of syllabic sound patterns, and constitute a reference time frame aligned on syllabic contours for interpreting gamma-based phonemic processing (*Shamir et al., 2009*; *Ghitza, 2011*; *Kayser et al., 2012*; *Panzeri et al., 2014*).

Compelling as this hypothesis may sound, direct evidence for neural mechanisms linking speech constituents and oscillatory components is still lacking. One way to address a causal role of oscillations in speech processing is computational modelling, as it permits to directly test the efficiency of cross-coupled theta and gamma oscillations as an instrument of speech de-multiplexing, parsing, and encoding. Previous models of speech processing involved only gamma oscillations in the context of isolated speech segments (*Shamir et al., 2009*) or did not involve neural oscillations at all (*Gütig and Sompolinsky, 2009*; *Yildiz et al., 2013*). On the other hand, previous models of cross-frequency coupled oscillations did not address sensory functions as parsing and de-multiplexing (*Jensen and Lisman, 1996*; *Tort et al., 2007*). Here, we examined how a biophysically inspired model of coupled theta and gamma neural oscillations can process continuous speech (spoken sentences). Specifically, we determined: (i) whether theta oscillations are able to accurately parse speech into syllables, (ii) whether syllable-related theta signal may serve as a reference time frame to improve gamma-based decoding of continuous speech; (iii) whether this decoding requires theta to modulate the activity of the gamma network. To address the last two points, we compared speech decoding performance of the model with two control versions of the network, in which we removed the neural connection entraining the theta neurons by speech fluctuations or the link that couples them to the gamma neurons.

## Results

### Model architecture and spontaneous behaviour

The model proposed here (*Figure 1A*) is inspired from cortical architecture (*Douglas and Martin, 2004*; *da Costa and Martin, 2010*) and function (*Lakatos et al., 2007*) as well as from previous biophysical models of cross-frequency coupled oscillation generation (*Tort et al., 2007*; *Kopell et al., 2010*; *Vierling-Claassen et al., 2010*). We used the well documented Pyramidal Interneuron Gamma (PING) model for implementing a gamma network: bursts of inhibitory neurons immediately follow bursts of excitatory neurons (*Jadi and Sejnowski, 2014*), creating the overall spiking rhythm. Given that gamma and theta oscillations are both locally present in superficial cortical layers (*Lakatos et al., 2005*), we assume similar local generation mechanisms for theta and gamma with a direct connection between them. Direct evidence for a local generation of theta oscillations in auditory cortex is still scarce (*Ainsworth et al., 2011*) and we cannot completely rule out that they might spread from remote generators (e.g., in the hippocampus; *Tort et al., 2007*; *Kopell et al., 2010*). Yet, we built the case for local generation from the following facts: (1) neocortical (somatosensory) theta oscillations are observed in vitro (*Fanselow et al., 2008*), (2) MEG, EEG, and combined EEG/FMRI recordings in humans show that theta activity phase-locks to speech amplitude envelope in A1 and immediate association cortex—but not beyond—(*Ahissar et al., 2001*; *Luo and Poeppel, 2007*; *Cogan and Poeppel, 2011*; *Morillon et al., 2012*), and (3) theta phase-locking to speech is not accompanied by power increase, arguing for a phase restructuring of a local oscillation (*Luo and Poeppel, 2007*). We assumed a similar generation mechanism for theta and gamma oscillations, with slower excitatory and inhibitory synaptic time constants for theta (*Kopell et al., 2010*; *Vierling-Claassen et al., 2010*). The distinct dynamics for the two modules reflect the diversity of inhibitory synaptic timescales observed experimentally, with Martinotti cells displaying slow synaptic inhibition ($Ti$ neurons), and basket cells showing faster inhibition decay ($Gi$ neurons) (*Silberberg and Markram, 2007*). We refer to the theta network as Pyramidal Interneuron Theta (PINTH), by analogy with PING. The full model is hence composed of a theta-generating module with interconnected spiking excitatory ($Te$) and inhibitory ($Ti$) neurons that spontaneously synchronize at theta frequency (6–8 Hz) through slow decaying inhibition; and of a gamma-generating module with excitatory ($Ge$) and inhibitory ($Gi$) neurons that burst at a faster rate (25–45 Hz) synchronized by fast decaying inhibition (PING; *Figure 1B*) (*Börgers and Kopell, 2005*). The firing pattern of our simulated neurons is sparse and weakly synchronous at rest, consistent with the low spiking rate of cortical neurons (*Brunel and Wang, 2003*) (*Figure 1—figure supplement 1D*). Unlike the classical 50–80 Hz PING seen in in vitro preparations of rat auditory cortex (*Ainsworth et al., 2011*), our network produced a lower gamma frequency around 30 Hz, as observed in human auditory cortex in response to speech (*Nourski et al., 2009*; *Pasley et al., 2012*).

At rest the PINTH population activity synchronizes at the theta timescale, and the PING population at the gamma time scale. Both the $Te$ and $Ge$ populations receive projections from a 'subcortical' module that mimics the nonlinear filtering of acoustic input by subcortical structures, which primarily includes a signal decomposition into 32 auditory channels (*Chi et al., 2005*). Individual excitatory neurons in the theta module received channel-averaged input while those in the gamma module received frequency selective input. Such a differential selectivity was motivated by experimental observations from intracranial recordings (*Morillon et al., 2012*; *Fontolan et al., 2014*) suggesting that unlike the gamma one, the theta response does not depend on the input spectrum. It also mirrors the dissociation in primate auditory cortex between a population of 'stereotyped' neurons responding very rapidly and non-selectively to any acoustic stimulus (putatively $Te$ neurons) and a population of 'modulated' neurons responding selectively to specific spectro-temporal features (putatively $Ge$ neurons) (*Brasselet et al., 2012*). Each $Ge$ neuron receives input from one specific channel, preserving the auditory tonotopy, so that the whole $Ge$ population represents the rich spectral structure of the stimulus. Each $Te$ neuron receives input from all the channels, i.e., the $Te$ population conveys a widely tuned temporal signal capturing slow stimulus fluctuations. Importantly, the two oscillating modules are connected through all-to-all connections from $Te$ neurons to $Ge$ neurons allowing the theta oscillations to control the activity of the faster gamma oscillations. This structure enables syllable boundary detection (through the theta module) to constrain the decoding of faster phonemic information. The output of the network is taken from the $Ge$ neurons as we assume that the $Ge$ neurons provide the input to higher-level cortical structures performing operations like phoneme

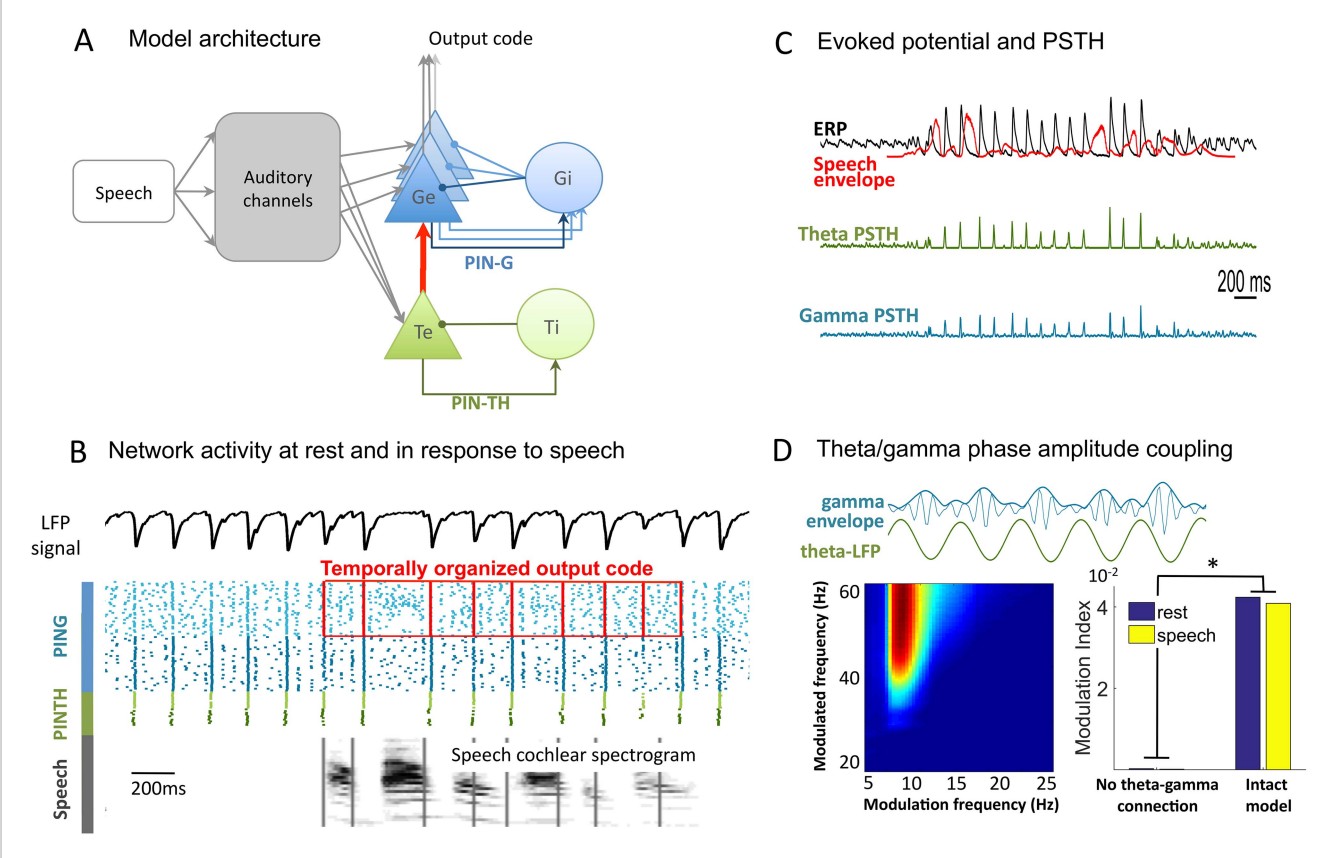

**Figure 1**. Network architecture and dynamics. (**A**) Architecture of the full model. *Te* excitatory neurons (n = 10) and *Ti* inhibitory neurons (n = 10) form the PINTH loop generating theta oscillations. *Ge* excitatory neurons (n = 32) and *Gi* inhibitory neurons (n = 32) form the PING loop generating gamma oscillations. *Te* neurons receive non-specific projections from all auditory channels, while *Ge* units receive specific projection from a single auditory channel, preserving tonotopy in the *Ge* population. PING and PINTH loops are coupled through all-to-all projections from *Te* to *Ge* units. (**B**) Network activity at rest and during speech perception. Raster plot of spikes from representative *Ti* (dark green), *Te* (light green), *Gi* (dark blue), and *Ge* (light blue). Simulated LFP is shown on top and the auditory spectrogram of the input sentence "*Ralph prepared red snapper with fresh lemon sauce for dinner*" is shown below. *Ge* spikes relative to theta burst (red boxes) form the output of the network. Gamma synchrony is visible in *Gi* spikes. (**C**) Evoked potential (ERP) and Post-stimulus time histograms (PSTH) of *Te* and *Ge* population from 50 simulations of the same sentence: ERP (i.e., simulated LFP averaged over simulations, black line), acoustic envelope of the sentence (red line, filtered at 20 Hz), PSTH for theta (green line) and gamma (blue line) neurons. Vertical bars show scale of 10 spikes for both PSTH. The theta network phase-locks to speech slow fluctuations and entrains the gamma network through the theta–gamma connection. (**D**) Theta/gamma phase-amplitude coupling in Ge spiking activity. Top panel: LFP gamma envelope follows LFP theta phase in single trials. Bottom‑Left panel: LFP phase-amplitude coupling (measured by Modulation Index) for pairs of frequencies during rest, showing peak in theta–gamma pairs. Bottom-right panel: MI phase-amplitude coupling at the spiking level for the intact model and a control model with no theta–gamma connection (red arrow on A panel), during rest (blue bars) and speech presentation (brown bars).

The following figure supplement is available for figure 1:

**Figure supplement 1**. Spectral analysis.

categorization and providing access to lexicon. Accordingly, in the model the *Ge* neurons receive more spectral details about speech than the *Te* neurons (*Figure 1B*). *Ge* spiking is then referenced with respect to timing of theta spikes, and submitted to decoding algorithms.

## Model dynamics in response to natural sentences

We first explored the dynamic behaviour of the model. As expected from its architecture and biophysical parameters (see 'Materials and methods'), the neural network produced activity in theta (6–8 Hz) and low gamma (25–45 Hz) ranges, both at rest and during speech presentation. Consistent with experimental observations (*Luo and Poeppel, 2007*) there was no notable increase in theta

spiking during speech presentation, but sentence onsets induced a phase-locking of theta oscillations as shown by the Post-stimulus time histograms of theta neurons, which was further enhanced by all edges in speech envelope. Consequently, the resulting global evoked activity followed the acoustic envelope of the speech signal (*Figure 1C*) (*Abrams et al., 2008*). Local Field Potential (LFP) indexes the global synaptic activity over the network (excitatory neurons of both networks) and its dynamics closely followed spiking dynamics. Unlike the LFP theta power pattern, the LFP theta phase pattern was robust across repetitions of the same sentence (*Figure 1—figure supplement 1A,C,E*), replicating LFP behaviour from the primate auditory cortex (*Kayser et al., 2009*), and human MEG data (*Luo and Poeppel, 2007*; *Luo et al., 2010*). In line with other empirical data from human auditory cortex (*Nourski et al., 2009*) gamma oscillations followed the onset of sentences (*Figure 1C*). Owing to the feed-forward connection from the theta to the gamma sub-circuits, the gamma amplitude was coupled to the theta phase both at rest and during speech (*Figure 1D*). The coupling was visible both in the spiking (*Figure 1—figure supplement 1B*) and LFP signal (*Figure 1D*). Critically, this coupling disappeared when the theta/gamma connection was removed, showing that a common input to *Te* and *Ge* cells is not sufficient to couple the two oscillations.

## Syllable boundary detection by theta oscillations

Before testing the speech decoding properties of the model, we explored whether syllable boundaries could reliably be detected at the cortical level by a theta network (see Methods). This first study was based on a corpus consisting of 4620 phonetically labelled English sentences (TIMIT *Linguistic Data Consortium, 1993*). The acoustic analysis of these sentences confirmed a correspondence between the dominant peak of the speech modulation spectrum and the mean syllabic rate (3–6 Hz) (*Figure 2—figure supplement 1A*), whereby syllabic boundaries correspond to trough in speech slow fluctuations (*Peelle et al., 2013*). The theta network in the model (*Figure 2—figure supplement 1B*) was explicitly designed to exploit such regularities and infer syllable boundaries. When presenting sentences to the theta module, we observed a consistent theta burst within 50 ms following syllable onset followed by a locking of theta oscillations to theta acoustic fluctuations in the speech signal (*Figure 2—figure supplement 1C,D*). More importantly, neuronal theta bursts closely aligned to the timing of syllable boundaries in the presented sentences (*Figure 2A*). We compared the performance of the theta network to that of two alternative models also susceptible to predict syllable boundaries: a simple linear-nonlinear acoustic boundary detector (*Figure 2—figure supplement 1E*) and Mermelstein algorithm, a state-of-the-art model which, unlike the model developed here, only permits 'off-line' syllable boundary detection (*Mermelstein, 1975*). The theta network performed better than both the linear model and the Mermelstein algorithm (*Figure 2B*, all p-values $<10^{-12}$). Similar to results from behavioural studies of human perception (*Miller et al., 1984*; *Nourski et al., 2009*; *Mukamel et al., 2011*) the theta network could adapt to different speech rates. The model performed better than other algorithms, with a syllabic alignment accuracy remaining well above chance levels (p < $10^{-12}$) in the twofold and threefold time compression conditions. (*Figure 2B*).

This first study demonstrates that theta activity provides a reliable, syllable-based, internal time reference that the neural system could use when reading out the activity of gamma neurons.

## Decoding of simple temporal stimuli from output spike patterns

Our next step was to test whether the theta-based syllable chunks of output spike trains (*Ge* neurons) for the different input types could be properly classified. We first quantified the model's ability to encode stimuli designed as simple temporal patterns. We used 50 ms sawtooth stimuli whose shape was parametrically varied by changing the peak position (*Figure 3A*), with interstimulus interval between 50 and 250 ms. This toy set of stimuli was previously used in a gamma-based speech encoding model and argued to represent idealized formant transitions (*Shamir et al., 2009*). We extracted spike patterns from all the *Ge* (output) neurons from −20 ms before each sawtooth onset to 20 ms after its offset. This procedure is referred to as 'stimulus timing' since it uses the stimulus onset as time reference. Using a clustering method (see 'Materials and methods'), we observed that the identity of the presented sawtooth could be decoded from the output spike patterns (*Figure 3A*) with over 60% accuracy (*Figure 3C*, light grey bar). We also computed the decoding performance when we used an internal time reference provided by the theta timing rather than by the stimulus timing. When spike patterns were analysed within a window defined by two successive theta bursts (*Figure 3C*, dark

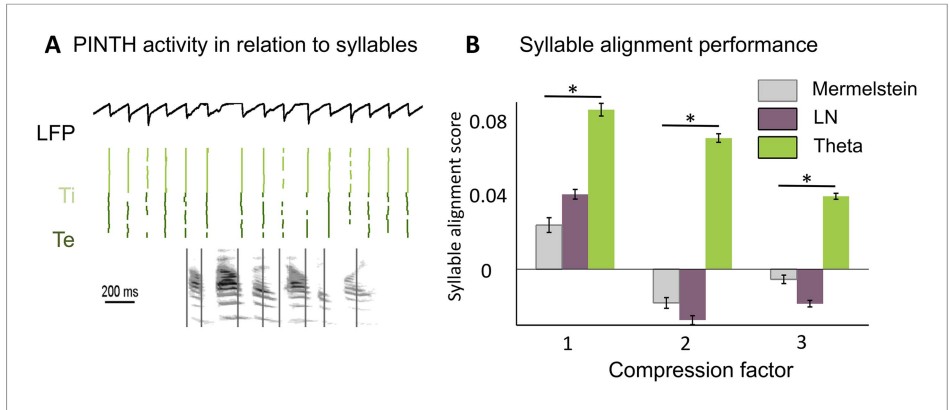

**Figure 2**. Theta entrainment by syllabic structure. (**A**) Theta spikes align to syllable boundaries. Top graph shows the activity of the theta network at rest and in response to a sentence, including the LFP traces displaying strong theta oscillations, and raster plots for spikes in the *Ti* (light green) and *Te* (dark green) populations. Theta bursts align well to the syllable boundaries obtained from labelled data (vertical black lines shown on top of auditory spectrogram in graph below). (**B**) Performance of different algorithms in predicting syllable onsets: Syllable alignment score indexes how well theta bursts aligned onto syllable boundaries for each sentence in the corpus, and the score was averaged over the 3620 sentences in the test data set (error bars: standard error). Results compare Mermelstein algorithm (grey bar), linear-nonlinear predictor (LN, pink) and theta network (green), both for normal speed speech (compression factor 1) and compressed speech (compression factors 2 and 3). Performance was assessed on a different subsample of sentences than those used for parameter fitting.
The following figure supplement is available for figure 2:

**Figure supplement 1**. TIMIT corpus and models used for syllable boundary detection.

grey bar), sawtooth decoding was still possible and even relatively well preserved (mean decoding rate of 41.7%). Noise in the theta module allows the alignment of theta bursts to stimulus onset and thus improves detection performance by enabling consistent theta chunking of spike patterns.

We then compared the decoding performance from the full model with that of two control models: one in which the theta module was not driven by the stimulus (*undriven theta* model) and one in which the theta module was not connected with the gamma module (*uncoupled theta/gamma* model) (*Figure 3B*, green and blue). Decoding performance of both control models, as revealed by the mean performance (*Figure 3C*) and confusion matrices (*Figure 3E*), was degraded for either neural code (theta onset and stimulus timing, all p-values $<10^{-9}$). The details of the raw confusion matrices show that the temporal patterns are decoded correctly or as a neighbouring temporal shape only in the intact version of the model (*Figure 3E*). Furthermore, the intact model achieved better signal vs rest discrimination than the two control models, notably avoiding false alarms (*Figure 3D*). In summary, these analyses show that gamma-spiking neurons within theta bursts provide a reliable internal code for characterizing simple temporal patterns, and that this ability is granted by the time-locking of theta neurons (*Te* units) to stimulus and the modulation they exert on the fast-scale output (*Ge*) units.

## Continuous speech encoding by model output spike patterns

The overarching goal of this theoretical work was to assess whether coupled cortical oscillations can achieve on-line speech decoding from *continuous* signal. We therefore set out to classify syllables from natural sentences. To decode *Ge* spiking, we used similar procedures as for the encoding/decoding of simple temporal patterns. Output *Ge* spikes were parsed into spike patterns based on the theta chunks, and the decoding analysis was used to recover syllable identity (*Figure 4A*). To evaluate the importance of the precise spike timing of gamma neurons, we compared decoding (see 'Materials and methods') using spike *patterns* (i.e., spikes labelled with their precise timing w.r.t. chunk onset) vs those obtained from plain spike *counts* (i.e., unlabelled spikes). When using spike patterns syllable decoding reached a high level of accuracy in the intact model: 58% of syllables were correctly classified within a set of 10 possible (randomly chosen) syllables (*Figure 4B*). Syllable

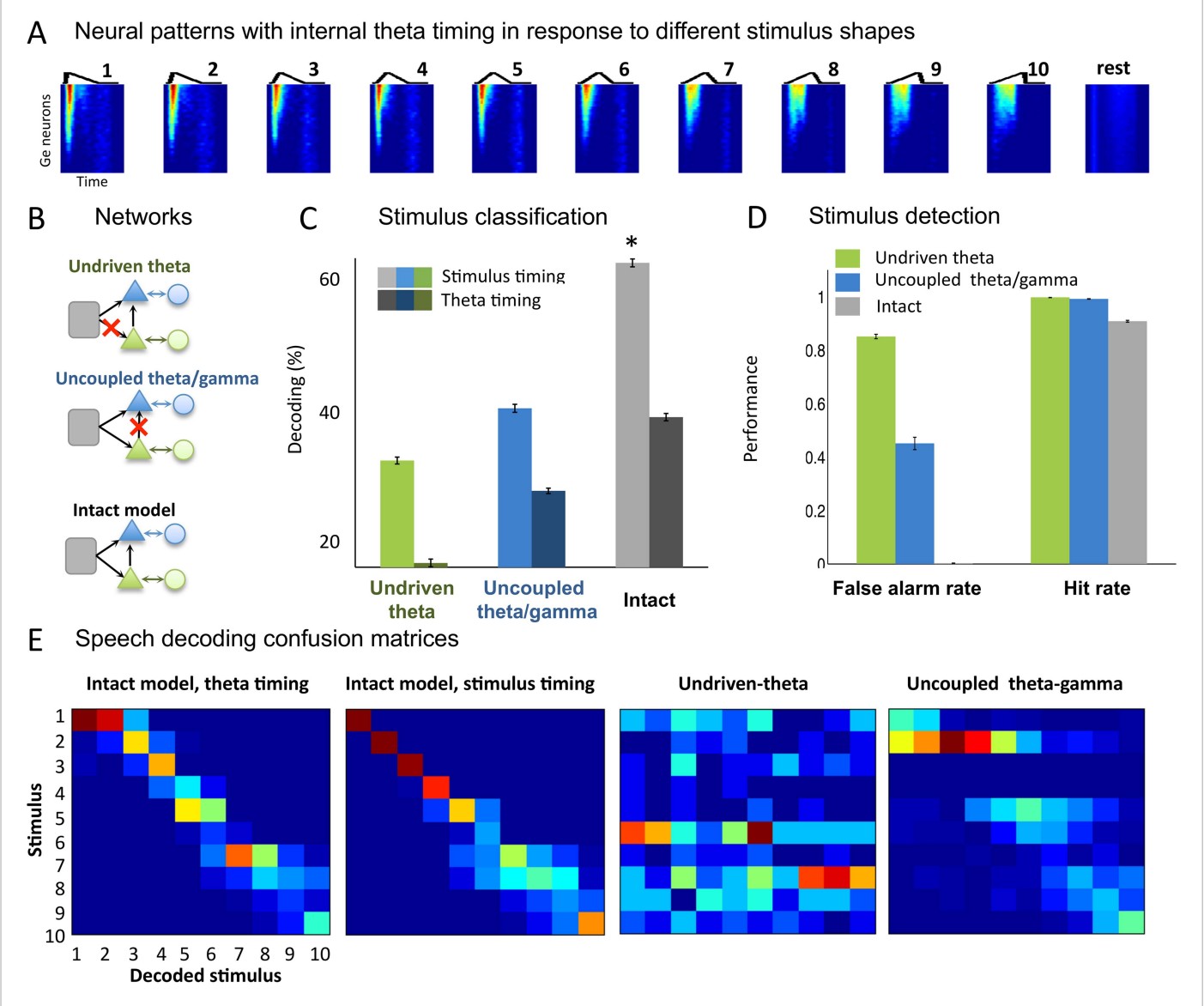

**Figure 3.** Sawtooth classification. (**A**) Gamma spiking patterns in response to simple stimuli. The model was presented with 50 ms sawtooth stimuli, where peak timing was parameterized between 0 (peak at onset) and 1 (peak at offset). Spiking is shown for different *Ge* neurons (y axis) in windows phase-locked to theta bursts (−20 to +70 ms around the burst, x-axis). Neural patterns are plotted below the corresponding sawtooths. (**B**) Simulated networks. The analysis was performed on simulated data from three distinct networks: 'Undriven-theta model' (no speech input to *Te* units, top), 'Uncoupled theta/gamma model' (no projection from *Te* to *Ge* units, middle), full intact model (bottom). (**C**) Classification performance using stimulus vs. theta timing for the three simulated networks. The stimulus timing (light bars) is obtained by extracting *Ge* spikes in a fixed-size window locked to the onset of the external stimulus; the theta timing (dark bars) is obtained by extracting *Ge* spikes in a window defined by consecutive theta bursts (*theta chunk*, see **Figure 3A**). Classification was repeated 10 times for each network and neural code, and mean values and standard deviation were extracted. Average expected chance level is 10%. (**D**) Stimulus detection performance, for the intact and control models. Rest neural patterns were discriminated against any of the 10 neural patterns defined by the 10 distinct temporal shapes. (**E**) Confusion matrices for stimulus- and theta-timing and the two control models (using theta-timing code). The colour of each cell represents the number of trials where a stimulus parameter was associated with a decoded parameter (blue: low numbers; red: high numbers). Values on the diagonal represent correct decoding.

decoding dropped when using spike counts instead of spike patterns (p < 10⁻¹²). Critically, decoding was poor in both control models (undriven theta and uncoupled theta/gamma) using either spike counts or spike patterns (significantly lower than decoding using spike patterns in the full model, all p-values < 10⁻¹², and non-significantly higher than decoding using spike counts in the full model, all p-values > 0.08 uncorrected).

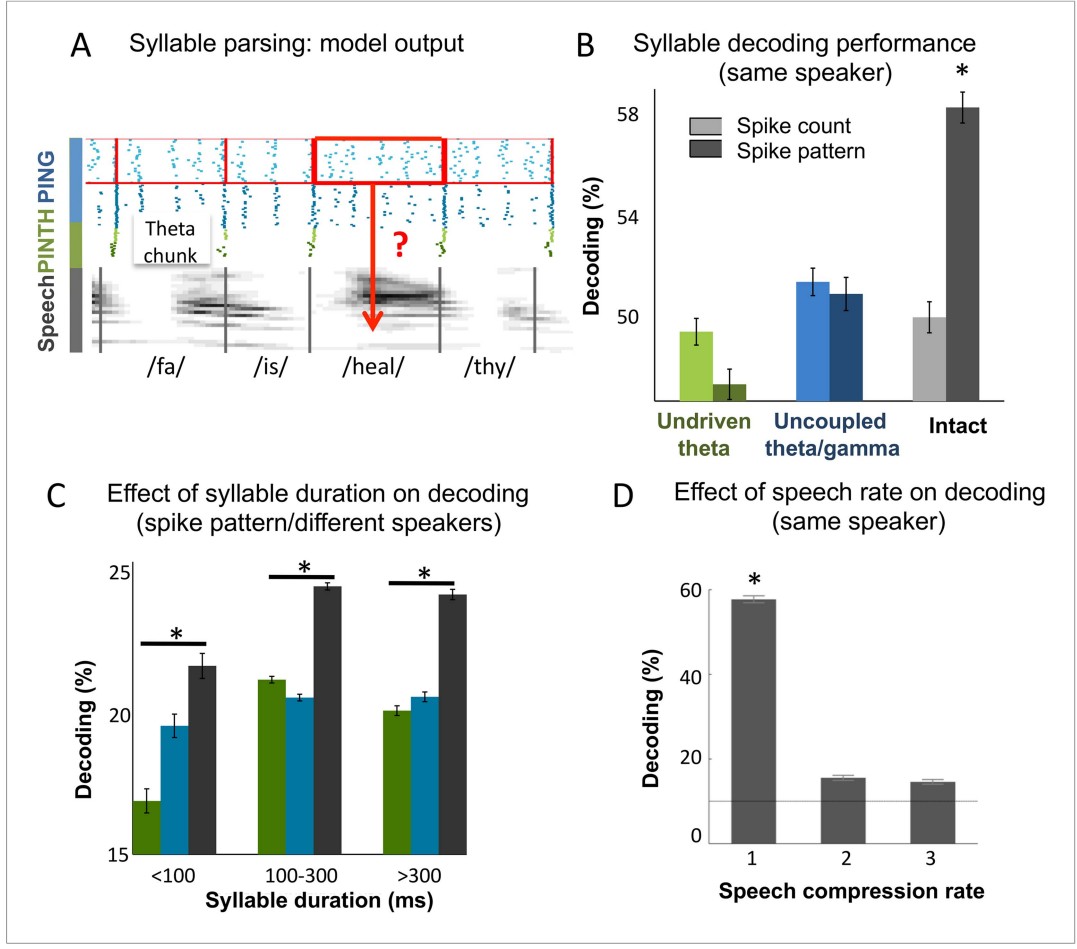

**Figure 4**. Continuous speech parsing and syllable classification. (**A**) Decoding scheme. Output spike patterns were built by extracting *Ge* spikes occurring within time windows defined by consecutive theta bursts (red boxes) during speech processing simulations. Each output pattern was then labelled with the corresponding syllable (grey bars). (**B**) Syllable decoding average performance for uncompressed speech. Performance for the three simulated models (*Figure 3B*) using two possible neural codes: *spike count* and *spike pattern*. (**C**) Syllable decoding average performance across speakers, using the spike pattern code. Syllable decoding was optimal when syllable duration was within the 100–300 ms range, i.e., corresponded to the duration of one theta cycle. The intact model performed better than the two controls irrespective of syllable duration range. Chance level is 10%. Colour code same as **B**. (**D**) Syllable decoding performance for compressed speech for the intact model using the spike pattern code (same speaker, as in **B**). Compression ranges from 1 (uncompressed) to 3. Average chance level is 10% (horizontal line in the right plot).

The following figure supplement is available for figure 4:

**Figure supplement 1**. Syllable classification across speakers.

We also explored the model performance for encoding syllables spoken by different speakers. We used a similar decoding procedure as above, but here the classifier was trained on different speakers pronouncing the same two sentences. Theta chunks were classified into syllables based on the network response to the two sentences uttered by 99 other speakers. The material included sentences spoken by 462 speakers of various ethnic and geographical origins, showing a marked heterogeneity in phonemic realization and syllable durations (as labelled by phoneticians). The syllable duration distribution was skewed with the median at 200 ms and tail values ranging from a few ms to over 800 ms (*Figure 4—figure supplement 1A*). Given that theta activity is meant to operate in a 3–9 Hz range, i.e., integrate speech chunks of about 100–300 ms (*Ghitza, 2011*, *2014*), we did not expect the model

to perform equally well along the whole syllable duration range. Accordingly, decoding accuracy was not uniform across the whole syllable duration range. When decoding from spike pattern, the intact model allowed 24% accuracy (chance level at 10%). It showed a peak in performance in the range in which it is expected to operate, i.e., for syllables durations between 100 and 300 ms. Given the cross-speaker phonemic variability such a performance is fairly good. Critically, the intact model outperformed control models both within the 100 to 300 ms range (p < 0.001), and throughout the whole syllable duration span (p < 0.001). These analyses overall show that the model can flexibly track syllables within a physiological operating window, and that syllable decoding relies on the integrity of the model architecture.

Lastly, we tested more directly the resilience of the spike pattern code to speech temporal compression and found that while degrading the decoding performance remained above chance for compression rates of 2 and 3 (*Figure 4D*), mimicking humans decoding performance (*Ahissar et al., 2001*). Altogether, the decoding of syllables from continuous speech showed that coupled theta and gamma oscillations provide a viable instrument for syllable parsing and decoding, and that its performance relies on the coupling between the two oscillation networks.

## Encoding properties of model neurons

We finally assessed the physiological plausibility of the model by comparing the encoding properties of the simulated neurons, without further parameter fitting, with those of neurons recorded from primate auditory cortex (*Kayser et al., 2009*; *2012*). The first analysis of neural encoding properties consisted of comparing the ability to classify neural codes from the model into arbitrary speech segments of fixed duration (as opposed to classification into syllables as in previous section). We simulated data using natural speech and studied the spiking activity of $Ge$ neurons by implementing the same methods of analysis as in the original experiment. We extracted fixed-size windows of spike patterns activity for individual $Ge$ neurons, and assessed neural encoding characteristics using different neural codes. Speech encoding was first evaluated using a nearest-mean classifier and then using mutual information techniques (*Kayser et al., 2009*).

### Classifier analysis

In this analysis, neural patterns were classified not into syllables as above or into any linguistic constituent but into arbitrary segments of speech, allowing for a-theoretical insight into the encoding properties of neurons. We extracted a subset of 25 sentences from the TIMIT corpus and exposed the network to 50 presentations of each sentence from the subset. We defined 10 stimuli as 10 distinct windows of a given size (from 80 to 480 ms) randomly extracted from the 25 sentences, and then assessed the capacity to decode the identity of a stimulus from the activity of individual $Ge$ neurons within that window (*Kayser et al., 2012*). Three different codes were used (*Figure 5A*): a simple *spike count* was used as reference code; a *time-partitioned code* where spikes were assigned to one of 8 bins of equal duration within the temporal window; a *phase-partitioned code* where spikes were labelled with the phase of LFP theta at the timing of spike (the spikes were then assigned into one of 8 bins according to their phase).

We observed that for 80 to 240 ms windows (within one theta cycle), decoding was almost as good for the phase-partitioned code as for the time-partitioned code (*Figure 5B*, left). In other words, stimulus decoding using theta timing was nearly as good as when using stimulus timing. Performance using the spike count was considerably lower (p < $10^{-12}$ for all 6 window sizes). Overall, there was a qualitative and even quantitative match between the results from simulated data and the original experimental results (*Figure 5B*, right). When we removed either the input-to-theta (undriven theta model) or the theta-to-gamma connection (uncoupled theta/gamma model) in the network, the performance of the phase-partitioned code dropped to just above that of the spike count code (*Figure 5—figure supplement 1A*; significantly lower increase in decoding performance using phase-partitioned instead of spike count code compared to full model, p < $10^{-12}$ for all 6 window sizes and both control models), and the simulations no longer predicted the experimental results. Finally, experimental data and simulations from the intact model also matched when we investigated the dependence of decoding accuracy on the number of bins, which was not the case for any of the control models (*Figure 5—figure supplement 1B*).

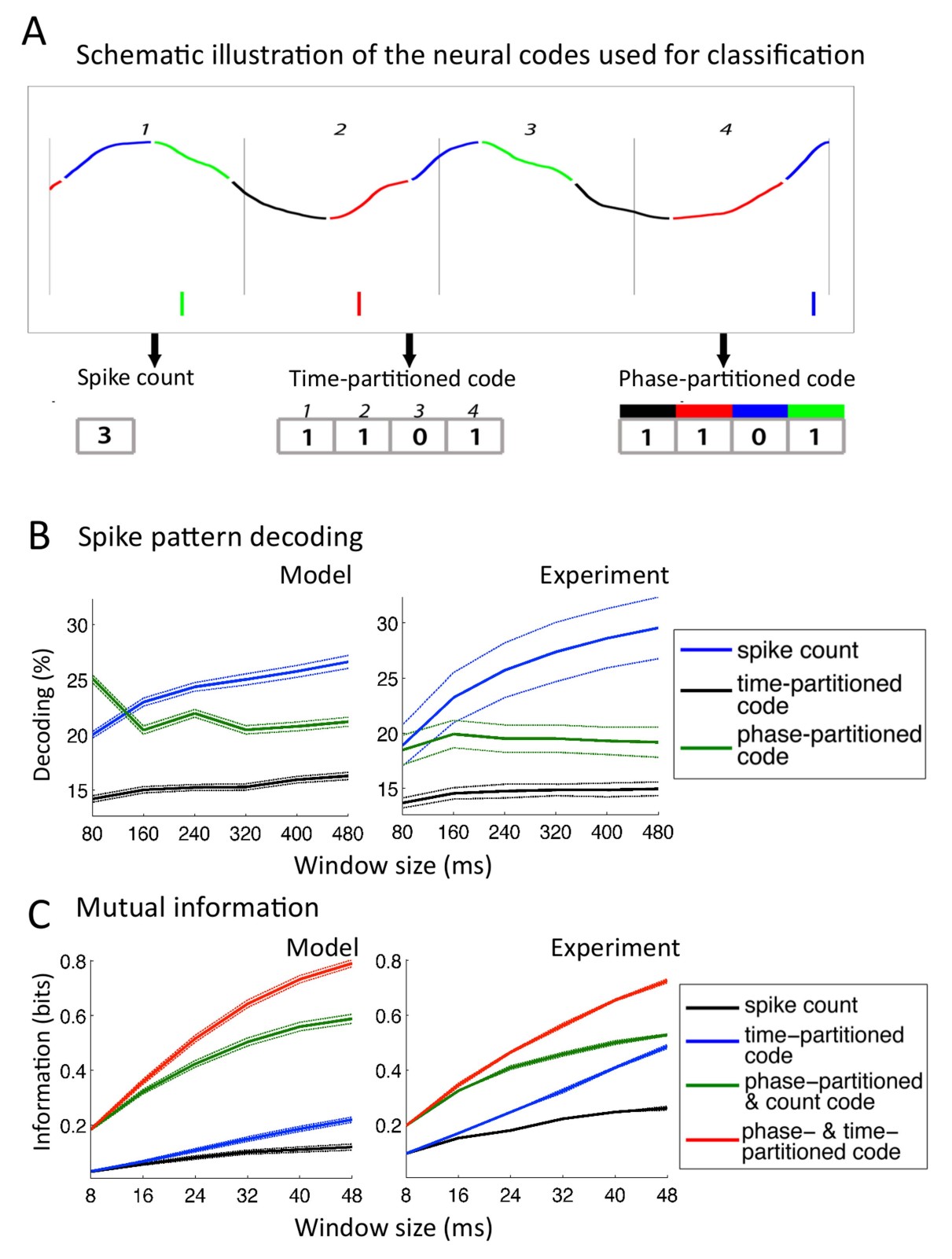

**Figure 5**. Comparison with encoding properties of auditory cortical neurons. (**A**) Neural codes. Stimulus decoding was performed on patterns of *Ge* spikes chunked in fixed-size windows (the figure illustrates the pattern for one neuron extracted from one window). *Spike count* consisted of counting all spikes for each neuron within the window. *Time-partitioned code* was obtained in dividing the window in *N* equal size bins (vertical grey bars) and counting spikes within each bin. *Phase-partitioned code* was obtained by binning LFP phase into *N* bins (depicted by the four colours in the top graph)
*Figure 5. continued on next page*

*Figure 5. Continued*

and assigning each spike with the corresponding phase bin. (**B**) Spike pattern decoding. (Left) Decoding performance across *Ge* neurons for the intact model using *N* = 8 bins for each code: spike count (black curve), time-partitioned (blue curve), and phase-partitioned codes (green curve). (Right) Data from the original experiment. Adapted from *Kayser et al., 2012*. (**C**) Mutual information (MI). (Left) Mean MI between stimulus and individual output neuron activity during sentence processing in the intact model for spike count (black curve), time-partitioned (blue line), combined count and phase-partitioned (green line) and combined time- and phase-partitioned codes (red line). (Right) Comparison with experimental data from auditory cortex neurons (adapted from *Kayser et al., 2009*).

The following figure supplement is available for figure 5:

**Figure supplement 1**. Speech decoding performance and MI (control models).

## Mutual information (MI) analysis

MI between the input (acoustic stimulus) and the output (neural pattern) provides an alternative measure for how well stimuli are encoded in the output pattern (see 'Materials and methods'). We used the same simulation data as for the classification procedure, but the sentences were subdivided into shorter chunks using a non-overlapping time window (length T: 8–48 ms) (*Kayser et al., 2009*). We compared the MI between the stimulus and neural activity in individual *Ge* neurons as a function of the length of stimulus window, using four neural codes: spike count, time-partitioned code, phase-partitioned code combined with spike count and finally combined phase- and time-partitioned codes. These codes are qualitatively equivalent to the decoding strategies used in the previous classifier analysis. *Figure 5C* shows that taking into account the spike phase boosts the MI carried by the *Spike count code* or the *Time-partitioned code* alone ($p < 10^{-12}$ for all 6 window sizes). In other words, spike phase provided additional rather than redundant information to more traditional codes. The gain provided by spike phase increased when enlarging the window and when combined with either spike count or spike pattern (*Spike Count* vs *Time-partitioned*; *Spike count and Phase-partitioned code* vs *Time- and Phase-partitioned code*). These results replicate the original experimental data from monkey auditory cortex (*Kayser et al., 2009*). Such a pattern was not reproduced using any of the control models (*Figure 5—figure supplement 1C*). These results hence show that in addition to enhancing the reliability of the spike phase code, the theta–gamma connection enhanced the temporal precision of *Ge* neurons spiking in response to speech stimuli.

Critically, results from both classifier and mutual information analyses demonstrate that the full network architecture of the model provides an efficient way of boosting the encoding capacity of neurons in a way that bears remarkable similarities to actual neurons from primate auditory cortex.

## Discussion

Like most complex natural patterns, speech contains rhythmic activity at different scales that conveys different and sometimes non-independent categories of information. Using a biophysically inspired model of auditory cortex function, we show that cortical theta–gamma cross-frequency coupling provides a means of using the timing of syllables to orchestrate the readout of speech-induced gamma activity. The current modelling data demonstrate that theta bursts generated by a theta (PINTH) network can predict 'on-line' syllable boundaries at least as accurately as state-of-the-art offline syllable detection algorithms. Syllable boundary detection by a theta network hence provides an endogenous time reference for speech decoding. Our simulated data further show that a gamma biophysical network, receiving a spectral decomposition of speech as input, can take advantage of the theta time reference to encode fast phonemic information. The central result of our work is that the gamma network could efficiently encode temporal patterns (from simple sawtooths to natural speech), as long as it was entrained by the theta rhythm driven by syllable boundaries. The proposed theta/gamma network displayed sophisticated spectral and encoding properties that compared both qualitatively and quantitatively to existing neurophysiological evidence including cross-frequency coupling properties (*Schroeder and Lakatos, 2009*) and theta-referenced stimulus encoding (*Kayser et al., 2009*; *2012*). The projections from the *Te* to *Ge* neurons endowed the network with phase-amplitude and phase-frequency coupling between gamma and theta oscillations, at both the spike and the LFP levels (*Jensen and Colgin, 2007*). This closely reproduces the theta/gamma

phase-amplitude coupling observed from intracortical recordings (*Giraud and Poeppel, 2012*; *Lakatos et al., 2005*). Importantly, due to the dissociation of excitatory populations we obtained denser gamma spiking immediately after the theta burst evoked by the syllable onset. This validates a critical point of theta/gamma parsing system, namely that a more in-depth encoding is carried-out by the auditory cortex during the early phase of syllables, when more information needs to be extracted (*Schroeder and Lakatos, 2009*; *Giraud and Poeppel, 2012*).

The human auditory system, like other sensory systems, is able to produce invariant responses to different physical presentations of the same input. Importantly, it is relatively insensitive to the speed at which speech is being produced. Speech can double in speed from one speaker to another and yet remain intelligible up to an artificial compression factor of 3 (*Ahissar et al., 2001*). In the current model, theta bursts could still signal syllable boundaries when speech was compressed by a factor 2 and this alignment deteriorated for higher compression factors. Syllable decoding was significantly degraded for compressed speech, yet remained twice as accurate as chance. Our network is purely bottom-up and does not include high level linguistic processes and representations, which in all likelihood plays an important role in speech perception (*Davis et al., 2011*; *Peelle et al., 2013*; *Gagnepain et al., 2012*): its relative resilience to speech compression is thus a fairly good performance. A previous model (*Gütig and Sompolinsky, 2009*) proposed a neural code that was robust to speech warping, based on the notion that individual neurons correct for speech rate by their overall level of activity. While this model achieved very good speech categorization performance, it relied on extremely precise spiking behaviour (neurons spiked only once, when their associated channel reached a certain threshold), for which neurophysiological evidence is scarce. Another model developed by Hopfield proposes that a low gamma external current provides encoding neurons with reliable timing and dynamical memory spanning up to 200 ms, a long enough window to integrate information over a full syllable (*Hopfield, 2004*). The utility of gamma oscillations for precise spiking is arguably similar in both Hopfield's model and ours, whereas the syllable integration process is irregularly ensured by intermittent traces of recent (~200 ms) neural activity in Hopfield's, and in ours by regularly spaced theta bursts that are locked to the speech signal. The advantage of our model is that integration over long speech segments is *permanently* enabled by the phase of output spikes with respect to the ongoing theta oscillation. Our approach shows that accurate encoding can be achieved using a system that does not require explicit memory processes, and in which the temporal integration buffer is only emulated by a slow neural oscillator aligned to speech dynamics.

In the current combined theta/gamma model, theta oscillations do not only act as a syllable-scale integration buffer, but also as a precise neural timer. Because syllabic contours are reflected in the slow modulations of speech, the theta oscillator can flexibly entrain to them (3–7 Hz, *Figure 2—figure supplement 1A*) and signal syllable boundaries. The spiking behaviour of theta neurons parallels experimental observations that a subset of neurons in A1 respond to the onset of naturalistic sounds (*Fishbach et al., 2001*; *Phillips et al., 2002*; *Wang et al., 2008*), providing an endogenous time reference that serves as a landmark to decode from other neurons (*Kayser et al., 2012*; *Brasselet et al., 2012*; *Panzeri and Diamond, 2010*; *Panzeri et al., 2014*). This parallels the dissociation between Ge and Te units in our model: while Ge units are channel specific, Te units cover the whole acoustic spectrum, which allow them to respond quickly and reliably to the onset of all auditory stimuli (*Brasselet et al., 2012*). In the model, however, theta neurons did not only discharge at stimulus onset but at regular landmarks along the speech signal, the syllable boundaries (*Zhou and Wang, 2010*). These neurons, hence, tie together the fast neural activity of gamma excitatory neurons into strings of linguistically relevant chunks (syllables), acting like punctuation in written language (*Lisman and Buzsáki, 2008*). This mechanism for segmentation is conceptually similar to the segmentation of neural codes by theta oscillations in the hippocampus during spatial navigation (*Gupta et al., 2012*).

From an evolutionary viewpoint, because the theta rhythm is neither auditory- nor human-specific, it might have been incorporated as a speech-parsing tool in the course of language evolution. Likewise, human language presumably optimized the length of its main constituents, syllables, to the parsing capacity of the auditory cortex. As a result, syllables have the ideal temporal format to interface with, e.g., hippocampal memory processes, or with motor routines reflecting other types of rhythmic mechanical constrains, e.g., the natural motion rate of the jaw (4Hz) (*Lieberman, 1985*).

Although conceptually promising, syllable tracking and speech encoding by a theta/gamma network, as proposed here, also show some limitations. While our current model is purely bottom-up, top-down predictions play a significant role in guiding speech perception (*Arnal and Giraud, 2012*;

*Gagnepain et al., 2012*; *Poeppel et al., 2008*) presumably across different frequency channels and processing timescales (*Wang, 2010*; *Bastos et al., 2012*; *Fontolan et al., 2014*). How these predictions interplay with theta- and gamma-parsing activity remain unclear (*Lee et al., 2013*). Experimental findings suggest that theta activity might be at the interface of bottom-up and top-down processes (*Peelle et al., 2013*). Theta auditory activity is better synchronized to speech modulations when speech is intelligible, irrespective of its temporal or spectral structure (*Luo and Poeppel, 2007*; *Peelle et al., 2013*). In the present model, theta activity bears an intrinsic temporal predictive function: it is driven by speech modulations, but is also resilient enough to syllable length variations to stay tuned to the global statistics of speech (average syllable duration). The model performed well above chance level when decoding syllables from a new speaker, showing flexibility in syllable tracking within a 3 to 9 Hz range. A natural follow-up of this work will hence be to explore how the intrinsic dynamics of theta and gamma activity interact not only with sensory input but also with linguistic top-down signals, e.g., word, sentence level predictions (*Gagnepain et al., 2012*), and even cross-modal predictions (*Arnal et al., 2009*). The trade-off between the autonomous functioning of theta and gamma oscillatory activity on one hand and their entrainment to sensory input on the other hand are at the core of future experimental and theoretical challenges.

In conclusion, our model provides a direct evidence that theta/gamma coupled oscillations can be a viable instrument to de-multiplex speech, and by extension to analyse complex sensory scenes at different timescales in parallel. By tying the gamma-organized spiking to the syllable boundaries, theta activity allows for decoding individual syllables in continuous speech streams. The model demonstrates the computational value of neural oscillations for parsing sensory stimuli based on their temporal properties and offers new perspectives for syllable-based automatic speech recognition (*Wu et al., 1997*) and brain-machine interfaces using oscillation-based neuromorphic algorithms.

## Materials and methods

### Architecture of the full model

The model is composed of 4 types of cells: theta inhibitory neurons (*Ti*, 10 neurons), theta excitatory cells (*Te*, 10 neurons), gamma inhibitory neurons (*Gi*, 32 neurons), and gamma excitatory neurons (*Ge*, 32 neurons) also called *output* neurons. All neurons were modeled as leaky integrate-and-fire neurons, where the dynamics of the membrane potential $V_i$ of the neurons followed:

$$C dV_i/dt = g_L(V_L - V_i) + I_i^{SYN}(t) + I_i^{INP}(t) + I_i^{DC} + \eta(t),$$

where $C$ is the capacitance of the membrane potential; $g_L$ and $V_L$ are the conductance and equilibrium potential of the leak current; $I^{SYN}$, $I^{INP}$ and $I^{DC}$ are the synaptic and constant currents, respectively; $\eta(t)$ is a Gaussian noise term of $\sigma_i$ variance.

Whenever $V_i$ reached the threshold potential $V_{THR}$, the neuron emitted a spike and $V_i$ was turned back to $V_{RESET}$.

$I^{SYN}$ is the sum of all synaptic currents from all projecting neurons in the network:

$$I_i^{SYN}(t) = \sum_j g_{ij} s_{ij}(t) \left( V_j^{SYN} - V_i(t) \right),$$

where $g_{ij}$ is the synaptic conductance of the *j-to-i* synapse, $s_{ij}(t)$ is the corresponding activation variable, and $V^{SYN}$ is the equilibrium potential of synaptic current (0 mV for excitatory neurons, −80 mV for inhibitory neurons). The activation variable $s_{ij}(t)$ varies as follow:

$$dx_j^R/dt = -1/\tau_j^R + \delta \left( t - t_j^{SPK} \right),$$

$$ds_{ij}/dt = -1/\tau_j^D,$$

where $\tau_j^R$ and $\tau_j^D$ are the time constants for synaptic rise and synaptic decay, respectively.

The connectivity among the cells is the following:

1. *Te* and *Ti* are reciprocally connected with all-to-all connections, generating the PINTH rhythm. There were also all-to-all connections within *Ti* cells.
2. *Ge* and *Gi* are also reciprocally connected with all-to-all connections, generating the PING rhythm.
3. *Te* projected with all-to-all connections to *Ge* cells, enabling cross-frequency coupling.

Input current $I_i^{INP}(t)$ is non-null only for Te and Ge cells and follows the equation:

$$I_i^{INP}(t) = \sum_c \omega_{ci} x_c(t),$$

where $x_c(t)$ is the signal from channel $c$ and $\omega_{ci}$ is the weight of the projection from channel $c$ to unit $i$.

Input to Te units is computed by filtering the auditory spectrogram by an optimized 2D spectro-temporal kernel (see section LN model below). LFP signal was simulated by summing the absolute values of all synaptic currents to all excitatory cells (both Ge and Te), as in *Mazzoni et al. (2008)*. All simulations were run on Matlab. Differential equations were solved using Euler method with a time step of 0.005 ms. Values for all parameters are provided in *Tables 1* and *2*.

## Stimuli

We used oral recordings of English sentences produced by male and female speakers from the TIMIT database (*Linguistic Data Consortium, 1993*). The sentences were first processed through a model of subcortical auditory processing (*Chi et al., 2005*) to the sentences. The model decomposes the auditory input into 128 channels of different frequency bands, reproducing the cochlear filterbank (http://www.isr.umd.edu/Labs/NSL/Software.htm). The frequency-decomposed signals undergo a series of nonlinear filters reflecting the computations taking place in the auditory nerve and other subcortical nuclei. We then reduced the number of channels from 128 to 32 by averaging the signal of each group of four consecutive channels, and used these 32 channels as input to the network. Each channel projected onto a distinct Ge cell (i.e., specific connections, $\omega_{ci} = 0.25\delta(c, i)$). As for Te input, each channel was convolved by the temporal filter and projected to all Te cells (all-to-all connections). Such a convolution can be implemented by a population of relay neurons that transmit their input with a certain delay, here between 0 and 50 ms.

Phoneme identity and boundaries have been labelled by phoneticians in every sentence of the corpus. We used the Tsylb2 program (*Fisher, 1996*) that automatically syllabifies phonetic transcriptions (*Kahn, 1976*) to merge these sequences of phonemes into sequences of syllables according to English grammar rules and thus get a timing for syllable boundaries.

To address the resilience of the model to speech compression, we produced compressed sentences by applying a pitch-synchronous, overlap and add (PSOLA) procedure implemented by PRAAT, a speech analysis and modification software (http://www.fon.hum.uva.nl/praat/). The procedure retains all spectral properties from the original speech data in the compressed process. The same precortical filters were then applied as for uncompressed data before feeding into the network.

## Syllable boundary prediction algorithms

Syllable boundaries triggered average (STAs) were computed as follow: for each syllable boundary (syllable onsets excluding the first of each sentence), we extracted a 700 ms window of the corresponding locked to the syllable boundary and averaged over all syllable boundaries. STAs were computed for speech envelope and for each channel of the *Chi et al. (2005)* model.

### Predictive models

We compared the performance of four distinct families of models to predict the timing of syllable boundaries based on speech envelope or speech audiogram: the Mermelstein algorithm, a Linear–Nonlinear (LN) model (a simplified integration-to-threshold algorithm), the entrained theta neural oscillator and a purely rhythmic control model. The four algorithms are presented in the sections below.

### Mermelstein algorithm

The Mermelstein algorithm is a standard algorithm that predicts syllable boundaries by identifying troughs in the power of the speech signal (*Mermelstein, 1975*; *Villing et al., 2004*). The predicted

---

**Table 1**. Full network parameter set

| Parameter | C | $V_{THR}$ | $V_{RESET}$ | $V_K$ | $V_L$ | $g_L$ | $g_{Ge,Gi}$ | $g_{Gi,Ge}$ | $g_{Te,Ge}$ |
|---|---|---|---|---|---|---|---|---|---|
| Value | 1 F/cm2 | −40 mV | −87 mV | −100 mV | −67 mV | 0.1 | 5/N_Ge | 5/N_Gi | 0.3/N_Te |

| Parameter | $\tau_{Ge}^R$ | $\tau_{Te}^R$ | $\tau_{Gi}^R$ | $\tau_{Ti}^R$ | $\tau_{Ge}^D$ | $\tau_{Gi}^D$ | $I_{Ge}^{DC}$ | $I_{Gi}^{DC}$ |
|---|---|---|---|---|---|---|---|---|
| Value | 0.2 ms | 4 ms | 0.5 ms | 5 ms | 2 ms | 20 ms | 3 | 1 |

**Table 2.** Optimal parameters for the LN model

| Parameter | $t_{sp}^{next}$ | $\tau_{lh}$ | DC |
|---|---|---|---|
| Value | 0.0748 | 1.433 | 0.4672 |

boundaries are computed according to the following steps. First, extract the power of speech signal in the 500–4000 Hz range (grossly corresponding to formants) and low-pass filter at 40 Hz to remove fast fluctuations, defining a so-called *loudness function*. Second, for each sentence, compute the convex hull of the loudness signal and extract the maximum of the difference between the loudness signal and its convex hull. If that difference exceeds a certain threshold $T_{min}$ and if the peak intensity of the interval of no more than $P_{max}$ smaller than the peak intensity of the whole sentence, then that time of maximal difference is defined as a predicted boundary and the same procedure is applied recursively to the intervals to the left and right of that boundary. Parameters $T_{min}$ and $P_{max}$ were optimized to yield minimum prediction distance (see below), yielding $T_{min} = 0.152\ dB$ and $P_{max} = 15.85\ dB$.

Note that this algorithm cannot be run *online* since the convex hull at a given time depends on the future value of speech power. Thus syllable boundaries can only be predicted after a certain delay, which makes it impractical for online speech comprehension as occurring in the human brain.

## LN model and variations

To evaluate the capacity of a simplified neural system to predict syllable boundaries, we trained a generalized linear point process model on the syllable data set. The model (*Figure 2—figure supplement 1D*) does not incorporate full neural dynamics but simply comprises a linear stimulus kernel followed by nonlinear function. The process issues a 'spike' or 'syllable boundary signal' whenever the output reaches a certain threshold (*Pillow et al., 2008*). This signal is fed back into the nonlinear function (another kernel *Ih* is used here): such negative feedback loop implements a relative refractory period. This model is a generalization of the Linear–Nonlinear Poisson model, hence we refer to it simply as *LN* model. We used the 32 auditory channels as input to the model and trained it to maximize its syllable boundary prediction performance.

We looked for a linear filter that is separable in its temporal and spectral component. We first computed the Spike Triggered Average (or rather 'Syllable Boundary Triggered Average') for all 32 channels from 600 ms to 0 ms prior to the actual boundary in 10 ms time steps. Yet *STA* provides the optimal estimate for the linear kernel in a LN model only when stimulus consists of uncorrelated white noise (*Chichilnisky, 2001*). To get the optimal values out of the white noise condition, we looked at the separable filter H that yields best prediction of the output, i.e., $(\langle |Y(t) - \hat{Y}(t|H)|^2 \rangle)$, where:

- $Y(t)$ is a binary output equal to 1 if there is a syllabic boundary in the 10 ms interval, 0 otherwise,
- $H$ is a separable spectro-temporal filter (i.e., $H(\omega, u) = S(\omega)T(u)$ for all orders u and all frequencies $\omega$. S and T are, respectively, the spectral and temporal component of filter H.
- $\hat{Y}(t|H) = \sum_{u,w} H(w, u) X(\omega, t - u)$, where $X(\omega, t)$ is the value of auditory channel $\omega$ at time step t.

Optimal solutions of the system verify:

$$\sum_u T(u)R(\omega, u) = \sum_{u,v,\xi} S(\xi)T(u)T(v)M(\omega, \xi, u, v)\ \ \forall \omega,$$
$$\sum_\omega S(\omega)R(\omega, u) = \sum_{v,\xi} S(\omega)S(\xi)T(v)M(\omega, \xi, u, v)\ \ \forall u,$$

where $R(\omega, u) = \langle Y(t)X(\omega, t) \rangle_t$ (i.e., R is the Spike Triggered Average) and M is the covariance tensor for X, i.e., $M(\omega, \xi, u, v) = cov(X(\omega, t - u), X(\xi, t - v))$.

Solutions to T and S for that system of equations can be approximated numerically using the following iterative procedure:

$$S_0(\omega) = 1\ \ \forall \omega, T_0(u) = 1\ \ \forall u,$$

$$S_{n+1} = \left( \frac{T_0 R}{\sum_{u,v} T_n(u)T_n(v)M(u,v,.,.)} \right)^T,$$

$$T_{n+1} = \left( \frac{R S_0}{\sum_{\omega,\xi} S_{n+1}(\omega)S_{n+1}(\xi)M(\omega, \xi, ., .)} \right),$$

and then stopping when the resulting square error $\|RS_0 - \sum_{\omega,\xi} S_{n+1}(\omega) S_{n+1}(\xi) T_n(v) M(\omega, \xi, ., .v)\|_u^2$ goes below a minimum value (we used a threshold of $10^{-4}$). The first 6 components (i.e., time bins) of the temporal kernel (i.e., 0–50 ms) were also used for input convolution in the theta model. We did not integrate further components (60–400 ms) since their weight was much lower and its implementation by relay neurons seemed less realistic.

To retrieve the optimal value for all parameters of the model, we used the GLM matlab toolbox developed in the Pillow lab (http://pillowlab.cps.utexas.edu/code_GLM.html), using as input the one-dimensional signal $U(t) = \sum_\omega S(\omega) X(\omega, t)$. Other parameters of the *LN* model including the self-inhibition temporal kernel *Ih* were optimized using the gradient descent implemented in the toolbox. This method provides estimation for a stochastic generalized *LN* model. We were interested in assessing the performance of a deterministic *LN* model. We then run a deterministic model with the same parameters as the stochastic model plus one new free parameter describing the normalized time to next spike (in the stochastic model, that time is drawn from an exponential distribution). The value of $t_{sp}^{next}$ was optimized using the same minimization procedure used for others models (see Optimisation section below). Two other parameters were also optimized again, since this procedure minimized a different score than the GLM toolbox score: time scale of self-inhibition $\tau_{Ih}$ and constant input to the model *DC* (**Table 2**).

We made one last modification to this *LN* model. We optimized the model such that it would maximally fire not at the time of syllable boundaries but 10 ms posterior to that time (*de facto*, we simply slid the STA window by 10 ms). This provides a delayed signal but likely more reliable since it can use more information (notably the rebound in the auditory spectrogram that is present right after a syllable boundary).

## Theta model

The theta model is composed of the *Te* and *Ti* cells from the full network model described above, with the exact same parameter set. 11 parameters were optimized in the full model, 10 in the control model (see values in **Table 3**).

## Control model

The control model was used to provide a baseline for assessing the performance of other models. Under these control conditions, predicted syllable boundaries were generated rhythmically at a fixed time interval, irrespective of the stimulus. The rate of the rhythmic process was varied from 1 Hz to 15 Hz in 0.5 Hz intervals. Such control model yielded better performance than another control model consisting of a homogeneous Poisson process. It thus provides a more stringent control for estimating the efficiency of other algorithms.

## Model performance evaluation

We evaluated how well syllable boundaries predicted by any model matched with the boundaries derived from labelled speech data. As an evaluation metrics, we used a point process distance that is used to compare distance between spike trains (**Victor and Purpura, 1997**). Shift cost was set to 20 s$^{-1}$ (in other words, a predicted and an actual boundary could be matched if they were no more than 50 msec apart).

To draw comparison between different models, for each level of compression, we computed the (non-normalized) distance measure for the theta model summed over all sentences in the test data set, as well as the average number of predicted boundaries per sentence. We then matched the theta model to a control rhythmic model with the same predicted syllabic rate, and computed the difference between the non-normalized distance for the theta model and for that matched rhythmic model.

## Optimisation

We optimized the parameters from all models to get the minimal normalized point process distance between predicted and actual boundaries in each sentence. Optimization was made using global

**Table 3**. Optimal parameters for the theta model

| Pars | $\sigma_{Te}$ | $\sigma_{Ti} = \sigma_{Ge} = \sigma_{Gi}$ | $\tau_{Te}^D$ | $\tau_{Ti}^D$ | $I_{Te}^{ext}$ | $I_{Te}^{DC}$ | $\tau_{Ti}^{DC}$ | $g_{Ti,Ti}$ | $g_{Ti,Te}$ | $g_{Te}^L$ |
|---|---|---|---|---|---|---|---|---|---|---|
| Value | 0.282 A $\sqrt{ms/cm^2}$ | 2.028 A $\sqrt{ms/cm^2}$ | 24.3 | 30.36 | 15 | 1.25 | 0.0851 | 0.432 | 0.207 | 0.264 |

gradient descent (function *fminsearch* in Matlab) and repeated with many initial points to avoid retaining a local minimum. Although both the theta model and the control model are intrinsically stochastic, the sample size was large enough for the objective function over the entire sample to be nearly deterministic, allowing for convergence of the gradient descent algorithm. The list of optimized parameters for each type of model is provided in the related model sections above. We split the entire TIMIT TRAIN data set (4620 sentences) into two data sets: a first data set of 1000 sentences was used to compute optimal parameters; final assessment of an algorithm performance with its optimal parameters was done on a separate set of 3620 sentences.

## Analysis of model behaviour

### LFP spectral analysis

Simulated LFP was downsampled to 1000 Hz before applying a time-frequency decomposition using complex Morlet wavelet transform, with all frequencies between 2 and 100 Hz with a 0.5 Hz precision. Coherence between stimulus and LFP signal was then computed for each time point *t* and each frequency *f* over 100 simulations using 100 distinct sentences *sen*, using the formula from *Mitra and Pesaran (1999)*. Synchronized bursts of the PING or PINTH were detected using spike timings in *Gi* and *Ti* populations since spikes of inhibitory neurons were more synchronized than those of excitatory neurons. Synchronous bursts of spikes were detected within a given population whenever more than 10% of neurons in the population spikes within a 6 ms interval (15 ms for *Ti* cells).

### Cross-frequency coupling

We computed cross-frequency coupling from 50 simulations of the model, each with a different TIMIT sentence preceded by 1000–1500 ms rest.

For the LFP phase-amplitude coupling, we extracted phase and amplitude from all frequencies from 2 Hz to 70 Hz in 1 Hz interval, and computed the Modulation Index for all pairs of frequencies (*Tort et al., 2010*). Data from all trials were concatenated (separately for spontaneous and speech-related activity) across all trials beforehand. To compute Modulation Index, in each condition, signal amplitude values $x(f_{amp},t,sen)$ were binned in $N = 18$ different bins according to the simultaneous phase of $x(f_{phase},t,sen)$. For spike phase-amplitude coupling, we defined spike gamma amplitude as the number of *Gi* neurons spiking at a given gamma burst, and the spike theta phase was defined by linear interpolation from $-\pi$ for a theta spike burst to $+\pi$ for the subsequent theta burst.

## Simple temporal patterns decoding

We first explored the model's performance using simple sawtooth signals (*Shamir et al., 2009*), representing prototypical realizations of formant transitions in a given frequency band. Each stimulus consisted of a rising component between 0 and 1, followed by a decay component from 1 back to 0. The overall length of the sawtooth was 50 ms, and the relative position of the maximal point $t_{MAX}$ between the starting point $t_{START}$ and end point $t_{END}$ was defined by a variable $a = (t_{MAX} - t_{START})/(t_{END} - t_{START})$.

The input connectivity had to be slightly modified since sawtooths are one-dimensional signals in contrast to the multi-dimensional channel signals that we have to use for speech stimuli: for *Te* units, we used $I_{Te}^{EXT} = 20$; and for the connections to *Ge* units in line with the original model (*Shamir et al., 2009*), we used different input levels across the population, ranging from 0.125 to 4 in 0.125 intervals. The rest of the model remained unchanged.

We simulated the response of the network to a series of 500 sawtooths with parameter *a* taking one of 10 equally spaced values within the [0 1] interval. Interstimulus interval varied randomly between 50 and 250 ms.

We compared the model's performance for different neural codes. For the 'stimulus timing' code (see 'Results' section), we extracted the spike pattern of output (*Ge*) neurons between 20 ms before and 70 ms after of each sawtooth onset. We computed the distance between all output spike patterns using a spike train distance measure (*Victor and Purpura, 1997*), implemented in the Spike Train Analysis Toolkit (http://neuroanalysis.org/toolkit/). We used a shift cost of 200 s$^{-1}$ corresponding to a timing resolution of 5 ms. We decoded the peak parameter using the simple leave-one-out clustering procedure of the STA toolkit, using a clustering exponent of −10. By comparing the 'decoded parameter', i.e., the parameter corresponding to the closest cluster, to the input sawtooth parameter, we built confusion matrices and computed decoding performance.

In the 'theta-timing' code, we extracted the spike pattern of output neuron in windows starting 20 before a theta burst and finishing 20 ms after the next theta burst ('*theta chunks*', *Figure 4A*). Spike times within each chunk were referenced with respect to the onset of the window. Each spike pattern was labelled with the corresponding value of the stimulus if the theta burst occurred during the presentation of the stimulus, or with the label 'rest' if the theta burst occurred during an interstimulus interval. The same decoding analysis was applied on such internally referenced neural patterns, yielding a 11 × 11 confusion matrix (10 stimulus shapes and rest). Detection theory measures (hits, misses, correct rejections, and false alarms) were computed by summing values in blocks of the confusion matrix (of size 10 × 10, 10 × 1, 1 × 10, and 1 × 1, respectively). A classification confusion matrix was obtained by removing the last row and last column of that confusion matrix.

We run the same decoding analysis on variants of the network: the full network; a control model where *Te* units do not receive the sawtooth input (*undriven theta network*) and another control where theta–gamma connections were removed (*uncoupled theta–gamma network*).

## Syllable decoding from sentences

The classification procedure was similar for syllable decoding, where we tried to decode the identity of syllables within continuous stream of speech (full sentences) from the activity of output neurons. We stimulated the network by presenting 25 sentences from the TIMIT corpus repeated 100 times each. We extracted theta chunks of *Ge* spike patterns as explained previously. Each chunk was labelled with the identity of the syllable being presented at the time of the first theta burst of the chunk. We randomly selected 10 syllables from the whole set of syllables within the 25 sentences. As in some cases there were several consecutive theta chunks corresponding to the same syllable, we equated the total number of theta chunks per syllable by randomly selecting 100 theta chunks labelled with each of the 10 syllables. Syllable classification of theta-chunked *Ge* spike patterns was performed using two different neural codes. For the *spike pattern code*, we applied the same procedure as for sawtooth classification, using a smaller value of spike shift cost corresponding to a timing resolution of 60 ms. For the *spike count code*, we measured the number of spikes emitted by each *Ge* neuron within a theta chunk. We then ran a simple nearest-mean classification procedure to decode syllable identity corresponding to each theta chunk from the spike counts of all *Ge* neurons (see 'Classification analysis' below). Both methods relied on the leave-one-out procedure that consists in identifying a chunk after the decoder was trained on all chunks but the to-be-decoded one. Decoding was repeated 200 times using each time a different set of 10 random syllables, and the analysis was performed over all three variants of the network.

For syllable classification across speakers, we used the two sentences from the TIMIT corpus that have been recorded for each of the 462 speakers ('*She had your dark suit in greasy wash water all year*' and '*Don't ask me to carry an oily rag like that*') and trained the network to classify syllables based on the neural output from other speakers, thus testing generalization across speakers. There is a wide variability of pronunciations over speakers as attested by the variability of chain of phonemes labelled of phoneticians, but the two sentences could nonetheless be parsed into 25 syllables overall for each speaker. We simulated the network presenting these 924 sentences and used the theta-chunked output to decode syllable identity. The method used was very similar to the syllable decoding analysis, where we classified theta-chunked neural patterns into one of 10 possible syllables (drawn randomly from the set of 25 syllables), with the only difference that here the classifier was based on theta chunks coming from different speakers. The classification was repeated 100 times for different subsets of syllables.

## Neural encoding properties: classification analysis

The first analysis of neural encoding properties consisted in comparing the ability to classify neural codes from the model into arbitrary speech segments (as opposed to syllables as in previous section). The methods, as detailed below, were inspired by the decoding of neural auditory cortical activity recorded in monkeys in response to naturalistic sounds (*Kayser et al., 2012*). We simulated the network by presenting 25 different sentences from the TIMIT corpus repeated 50 times each. For a given window size (ranging from 80 to 480 ms in 80 ms intervals), we randomly extracted 10 windows (defined as *stimuli*) from the overall set of 25 sentences. We then retrieved stimulus identity based on the activity of a neuron that was randomly drawn from the *Ge* population using three

different neural codes. In the *neural count code*, we counted the number of spikes emitted by that neuron within each window. In the *time-partitioned code*, we divided each window into *N* equally size bins, and computed the number of spikes for each of the 8 bins separately. In the *phase-partitioned code*, we divided the window based on theta-phase- rather time-intervals: each spike was labelled with the phase of the theta oscillation at the corresponding spike time, and we computed the number of spikes falling into each of the *N* subdivisions of the $[-\pi;\pi]$ interval.

We then used a nearest-mean template matching procedure to decode the stimuli. To classify each stimulus exemplar using each neural code, we averaged the vectors over all presentations of each stimulus using a leave-one-out procedure; we then computed the Euclidian distance from the current vector to each of the 10 stimulus-averaged template. Finally, we 'decoded' the neural code by assigning it to the stimulus class with minimal distance to template. A more detailed explanation of the procedure is provided in the original experiment article (*Kayser et al., 2012*). The procedure was repeated 1000 times, each time with a different set of 10 random stimuli, and performed the 3 variants of network.

## Neural encoding properties: mutual information analysis

We complemented the stimulus classification with a similar analysis using mutual information between the acoustic 'stimulus' and response of individual *Ge* neurons to further characterize the encoding properties of the network. Mutual Information (*MI*) estimates the reduction of uncertainty about the acoustic '*stimulus*' that is obtained from the knowledge of a single trial of neural response. The data set was identical to the one previously used for stimulus classification analysis, where each stimulus was again segmented into non-overlapping windows of length T (here 8 to 48 ms) (*Kayser et al., 2009*; *de Ruyter van Steveninck et al., Strong, 1997*).

Mutual Information was computed for the same neural codes as in *Kayser et al. (2009)*. We used *Spike count code* and *Time-partitioned code* as described above (for the Time-partitioned code the size of the bins was kept constant to 8 bins; the number of bins in a window hence increased with window size. As slow LFP phase was more reliable over sentence repetitions than power, we combined spike count and LFP theta phase to get a *Spike count & Phase-partitioned code* (*Montemurro et al., 2008*). For this code, the phase of slow LFP was divided into $N = 4$ bins, and the firing rate in each window was labelled according to the phase at which the first spike occurred. Finally, we explored the influence of slow LFP phase on MI when combined with temporal spiking patterns. Thus, in the *Time- & Phase-partitioned code* spikes carry two distinct tags, the first one referring to the position of the spike inside one of the four subdivisions of the stimulus window, the second indicating the phase of the underlying LFP at the moment of the spike occurrence.

We corrected for sampling bias (*Kayser et al., 2009*) first by using a *shuffling* method (*Panzeri et al., 2007*), then the quadratic extrapolation method (*Strong et al., 1998*). We further reduced the residual bias using a bootstrapping technique (200 resampled data) (*Montemurro et al., 2008*).

## Acknowledgements

This work was funded by the European Research Council (Compuslang project; Grant agreement 260347), the Swiss National Fund (grant 320030-149319), the Agence National de la Recherche, the CNRS. We warmly thank Oded Ghitza for stimulating discussions, Maoz Shamir and Andy Brughera for sharing elements of code with us, Adrien Wohrer for help with the mathematical analysis and Jean-Paul Haton for his input from the perspective of automatic speech recognition. BSG gratefully acknowledges partial support from the National Research University Higher School of Economics.

## Additional information

### Funding

| Funder | Grant reference | Author |
| --- | --- | --- |
| European Research Council (ERC) | CompusLang 260347 | Lorenzo Fontolan |
| Schweizerische Nationalfonds zur Förderung der Wissenschaftlichen Forschung | 320030-149319 | Anne-Lise Giraud |

| Funder | Grant reference | Author |
| --- | --- | --- |
| Agence Nationale de la Recherche | | Boris Gutkin |
| Centre National de la Recherche Scientifique | | Anne-Lise Giraud |

The funders had no role in study design, data collection and interpretation, or the decision to submit the work for publication.

## Author contributions

AH, Conception and design, Acquisition of data, Analysis and interpretation of data, Drafting or revising the article; LF, Acquisition of data, Analysis and interpretation of data, Drafting or revising the article; CK, Acquisition of data, Analysis and interpretation of data; BG, A-LG, Conception and design, Analysis and interpretation of data, Drafting or revising the article

## Additional files

### Major dataset

The following previously published dataset was used:

| Author(s) | Year | Dataset title | Dataset ID and/or URL | Database, license, and accessibility information |
| --- | --- | --- | --- | --- |
| Linguistic Data Consortium (LDC) | 1993 | TIMIT Acoustic-Phonetic Continuous Speech Corpus | https://catalog.ldc.upenn.edu/LDC93S1 | Available from the Linguistic Data Consortium (registration required). |

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
