## [Decision Letter]

Thank you for sending your work entitled "Speech encoding by coupled cortical theta and gamma oscillations" for consideration at *eLife*. Your article has been favorably evaluated by Eve Marder (Senior editor), three reviewers, and a member of our Board of Reviewing Editors (Hiram Brownell).

The Reviewing editor and the reviewers discussed their comments before we reached this decision, and the Reviewing editor has assembled the following comments to help you prepare a revised submission.

The study breaks new ground in that it shows that a simple, biologically plausible model of coupled theta and gamma oscillators performs better at speech decoding than other methods, including the authors' own circuit with uncoupled oscillators. While the reviewers value the potential contribution of this work, they were clear that they felt unable to fairly evaluate the suitability of the paper for publication without substantially more information. The major points to be addressed in a revision are presented below. Please note that presenting the additional information will not necessarily make the paper suitable for publication. With the requested additional information (e.g., results of statistical tests) presented as part of a revised submission, the reviewers will carry out a new review and arrive at a decision.

1) A general area of concern is the presentation of the model and the motivation for its specific architecture. Much of the *eLife* readership will not be familiar with the modeling literature.

1A) One specific issue is the motivation for the differential connectivity of theta- and gamma-associated neurons to the input signal).

1B) In the subsection headed “Model architecture and spontaneous behaviour” some of the aspects of the model appear somewhat ad-hoc: how realistic is it to assume that gamma and theta are generated by the same mechanism? How realistic is the connectivity structure between neurons? To preempt any concerns in this direction, one suggestion is to state explicitly and together the ways in which the model is based on neural data (connectivity structure, time constants, etc.) and the ways it is not. Some of this content is already stated in various places.

2) The connection of this work to prior literature could be improved.

2A) Given previous work on coupled oscillators and entrainment of oscillators, it should be made more clear what was known before the current work carried out and what is different and surprising about the results of the current work.

2B) The two main aims of the study could be better developed to set the stage for the rest of the manuscript. The first one (that speech constituents can be tracked with oscillations) has been shown with real data many times. A suggested rephrasing could include the modeling aspect. The second aim could be stated more clearly. (It is unclear what is meant by 'shape efficient neural code'.)

3) The description of the stimulus items used for training and testing the model is a major aspect of the paper needing additional detail. How do stimuli differ from each other? Since the test of decoding is a comparison of neural activity in response to test stimuli compared to that activity in response to trained stimuli, it is essential to know how different were the test and trained stimuli. In the subsection headed “Syllable decoding”, "25 sentences… repeated 100 times each" suggest identical stimuli were used. Was each sentence spoken by a different speaker? If not, especially if some test stimuli were identical to training stimuli, then trial-to-trial differences could be due to only internal noise in the circuit, which could always be lowered to improve performance in any circuit. Thus, relevant questions include: were different exemplars of syllables spoken by different speakers? Or did they appear in different phrases or words but spoken by the same speaker?

4) More information is needed regarding support for the model.

4A) It was not clear what data supported the network architecture, specifically, that the *Ge* neurons receive input from only one specific auditory channel, whereas the *Te* neurons receive input from all channels. Are the theta- and gamma-associated neurons anatomically dissociable?

4B) A major issue raised consistently in the reviews is that throughout the paper, the specific statistical tests applied were opaque or missing. Examples include p values (or Bayes factors, etc.) to back up many of the comparisons across conditions, syllable boundary detection (is the presented model's performance significantly better than the others?), and decoding of simple stimuli (again, is performance significantly better for the intact model?). What is the chance level of decoding for the conditions shown in Figure 3 and Figure 4? These are not necessarily what might be expected, depending on the number of samples available. Although the approaches seem generally reasonable, statistical support must be reported (incorporating, if applicable, correction for multiple comparisons).

5) Some additional comment on phase-reset is warranted. For example, Figure 2 nicely shows PINTH activity (LFP and theta-associated activity) occurring in a (quasi-)regular fashion before the presentation of the speech stimulus. The degree to which the incoming speech signal systematically alters the phase of this activity is unclear: from Figure 2, it appears to happen instantaneously, with no missed/extra spikes. Such performance seems intriguing but potentially unrealistic. How does the phase following speech presentation compare with that prior to speech presentation?

6) Please check figures and figure captions. For example, there are captions for Figure 2 C, D and E, but these panels are missing in Figure 2. (Note also a similar mismatch for Figure 2—figure supplement 1.)

[Editors' note: further revisions were requested prior to acceptance, as described below.]

Thank you for submitting your work entitled "Speech encoding by coupled cortical theta and gamma oscillations" for peer review at *eLife*. Your submission has been favorably evaluated by Eve Marder (Senior editor), a Reviewing editor, and three reviewers.

The reviewers have discussed the reviews with one another and the Reviewing editor has drafted this decision to help you prepare a revised submission.

Summary: The paper is greatly improved and needs only relatively small revisions prior to publication.

Essential revisions:

1) Please address the following in the paper itself. The comment is in response to a point made in the authors' cover letter.

A system without noise would always perform at 100% on any classification test if given repeats of certain items in training, and then tested on those same items. Your point that noise is needed to optimize detection of stimulus onset time is important and should be included in the paper (that is, for tests in which stimulus onset time is not provided to the model).

2) Given the authors' intent to take into account the constraint of neural noise, additional information as to how well this is done should be provided. At a minimum, in the spiking models, please provide the CV of the spike trains and the Fano factors for responses to identical stimuli. For the LFP signals, also provide an indicator of the trial-to-trial variability in response to identical stimuli: e.g., when stimuli are identical, CV of heights of particular LFP peaks, or of time intervals between specific peaks and troughs.

---

## [Author Response]

*1) A general area of concern is the presentation of the model and the motivation for its specific architecture. Much of the eLife readership will not be familiar with the modeling literature*.

*1A) One specific issue is the motivation for the differential connectivity of theta- and gamma-associated neurons to the input signal)*.

We thank the reviewers for bringing up this important point. Our rationale was as follows. Human intracortical data obtained in primary auditory cortex (53) indicate that the cerebro/acoustic coherence in the theta range does not depend on the input frequency. In the gamma range, however, coherence is stronger at specific stimulus frequencies (see Figure 6). This suggests that theta oscillations inside auditory cortex are broadly responsive to the whole hearing spectrum, while local gamma generators are finely tuned to specific frequencies. This claim also makes sense from the point of view of multiplexing to track a broadband and finely tuned signals simultaneously at two distinct frequencies (Panzeri et al. 2010; [26]). This was our main motivation for feeding a broadband signal to individual theta neurons and a more fine-grained, spectrally complex input to individual gamma neurons. This is now specified in the Results section of the manuscript together with the information requested in Point 3:

"Such a differential selectivity was motivated experimental observations from intracranial recordings (21; 53) suggesting that unlike the gamma one, the theta response does not depend on the input spectrum".

Author response image 1.Human stereotactic EEG recording in auditory cortex.A: the location of the electrode shaft, B: acoustic stimulus (one sentence); C: time-frequency representation of the cortical response in primary auditory cortex; D: spectrum of the stimulus in relation with E: the cross-correlation between stimulus (Hilbert transform of B) and cortical response for 33 different frequency bands (corresponding to 33 cochlear filters) spanning the speech audio spectrum up to 8 kHz. Note that the cross-correlation is broad-band in the theta range, but frequency specific in the gamma range, with peaks of correlation corresponding roughly to the energy peaks (speech formants). The data partly published in [53] and [25], and partly original.**DOI:**
http://dx.doi.org/10.7554/eLife.06213.017

*1B) In the subsection headed “Model architecture and spontaneous behaviour” some of the aspects of the model appear somewhat ad-hoc: how realistic is it to assume that gamma and theta are generated by the same mechanism? How realistic is the connectivity structure between neurons? To preempt any concerns in this direction, one suggestion is to state explicitly and together the ways in which the model is based on neural data (connectivity structure, time constants, etc.) and the ways it is not. Some of this content is already stated in various places*.

Thanks you for this suggestion. We have now updated the manuscript and put all the information regarding experimental support for the model architecture in the "Model architecture and spontaneous behaviour" subsection of the Results section. We believe that this change, motivated by reviewers comments, permit to better disentangle those assumptions that were directly constrained by experimental evidence, and those that were put forward for future experimental verification.

Regarding the generation of theta oscillations, we state (now in the section “Model architecture and spontaneous behaviour”) that in the absence of conclusive evidence about the underlying mechanisms, we chose the most parsimonious option: a mechanism similar to gamma generation (PING) based on the attested presence of inhibitory neurons with slower time constants (our *Ti* neurons), which were involved in theta generation in previous models of neocortical oscillations ([75]; Compte et al., J Neuroscience 2008).

When developing our model, we did consider alternative architectures but concluded that they were not as compelling. Most obvious example is a three-population model with *Gi* units*, Ti* units and a single population of pyramidal neurons, (as in Tort et al. PNAS 2005 model of hippocampus) was not appropriate as it did not permit a differential frequency channel tuning for the inputs to the gamma excitatory and the theta excitatory neurons (since they represent the same population). We decided to avoid a lengthy discussion about alternative models in the manuscript, as it would probably sound unappealing to most readers. Overall we believe these changes should add to the clarity of the manuscript.

*2) The connection of this work to prior literature could be improved*.

*2A) Given previous work on coupled oscillators and entrainment of oscillators, it should be made more clear what was known before the current work carried out and what is different and surprising about the results of the current work*.

We agree with this remark and have now thoroughly rewritten the Introduction to make a more explicit connection between the previous literature and the present study. To the best of our knowledge, until now coupled cross-frequency oscillations were only used to model the functions of the hippocampus and prefrontal cortex ([33]; Tort et al. 2005; etc.). These models were not envisaged to account for the sampling function of sensory areas. In particular, the functional interaction of an intrinsic oscillation with a sensory signal bearing pseudo-rhythmic modulations such as speech has, to the best of our knowledge, never been explored before. While supported by some experimental evidence, the role of coupled oscillations in speech processing has only been formulated from a neurophysiological perspective (25) or in phenomenological terms (Ghitza, Front Psychology 2010).

*2B) The two main aims of the study could be better developed to set the stage for the rest of the manuscript. The first one (that speech constituents can be tracked with oscillations) has been shown with real data many times. A suggested rephrasing could include the modeling aspect. The second aim could be stated more clearly. (It is unclear what is meant by 'shape efficient neural code'*.*)*

We respectfully disagree with the statement about our first aim. A number of experiments shows that neural oscillations in auditory cortex track the slow fluctuations of speech, but none of them has explicitly tied oscillations to specific linguistic constituents, such as we do here with respect to theta oscillations and syllables. Thus, the efficiency of biologically plausible coupled theta and gamma oscillators in speech encoding has never been evaluated before. Also, given the large variability of the syllabic rhythm in natural speech, it was not clear that neural oscillations could accurately track the syllabic rate. Finally, no experimental data has ever proven causality with respect to the hypothesized functions. It is precisely because causality cannot easily be established in human recordings (it would require a specific interference with the theta rhythm with e.g. optogenetics), that we initiated this modelling work.

The new Introduction paragraph now clarifies the current state of knowledge, and specifies the three specific aims of the study. Note that for the readability of the manuscript, the changes are not tracked in the Introduction.

*3) The description of the stimulus items used for training and testing the model is a major aspect of the paper needing additional detail. How do stimuli differ from each other? Since the test of decoding is a comparison of neural activity in response to test stimuli compared to that activity in response to trained stimuli, it is essential to know how different were the test and trained stimuli. In the subsection headed “Syllable decoding”, "25 sentences…repeated 100 times each" suggest identical stimuli were used. Was each sentence spoken by a different speaker? If not, especially if some test stimuli were identical to training stimuli, then trial-to-trial differences could be due to only internal noise in the circuit, which could always be lowered to improve performance in any circuit. Thus, relevant questions include: were different exemplars of syllables spoken by different speakers? Or did they appear in different phrases or words but spoken by the same speaker*?

As correctly pointed out by the reviewers, we used identical stimuli for syllable classification in the analysis where we aimed at comparing both different model versions and different neural codes. To thoroughly assess the different versions of the model and explore the added value of reading out gamma spiking as a function of theta phase, some consistency in syllable length was required. We believe that the differential results are both valid and important. To address the reviewers’ concern, we now additionally present decoding results when the classifier was trained on 2 sentences uttered by 462 different speakers, where the training material was hence different than the tested material. Despite the fact that the new dataset involved a wide range of phonemic realizations (as labelled by phoneticians) due to accents and pronunciation variants, classification reached 20-24% (depending on model version), which remained well above chance level (10%). Decoding accuracy using the intact model was significantly better than either control model, and importantly the full model performed optimally when the syllable duration was within the 100-300 ms range, i.e. roughly one theta cycle.

Details about this control analysis have been included in the Results section and we added a corresponding figure (Figure 4 and Figure 4—figure supplement 1).

The point about reducing noise in the system makes sense from an algorithmic point of view but less so from a neuroscience perspective. Neuronal noise is a neural constraint requiring robust computations. Moreover, if reducing noise in the gamma network may arguably enhance chance performance, it should be noted that the level of noise in the theta module was optimized for syllabic boundary detection, thus noisy theta is expected to allow better classification performance than noiseless theta.

We thank the reviewers for this remark and we believe the across-speaker decoding results contribute to strengthen the main claims of the manuscript.

*4) More information is needed regarding support for the model*.

*4A) It was not clear what data supported the network architecture – specifically, that the* Ge *neurons receive input from only one specific auditory channel, whereas the* Te *neurons receive input from all channels. Are the theta- and gamma-associated neurons anatomically dissociable*?

Regarding the different patterns of input to the *Te* and *Ge* populations, please refer to our reply to point 1A. Regarding the issue of anatomic dissociability, there is unfortunately very little experimental evidence to build from. We draw a putative link between our two subpopulations of theta and gamma neurons and the two subclasses of stereotyped and modulated neurons found in primate auditory cortex by Brasselet and colleagues (8). The authors report the existence of two subclasses of neurons in primate auditory cortex corresponding to our two populations of theta and gamma neurons (see Discussion):

A subclass of 'stereotyped' neurons responding very rapidly and non-selectively to any acoustic stimulus, presumably receiving input from the whole acoustic spectrum, which we model as theta neurons, signalling boundaries in the speech signal.

A subclass of 'modulated' neurons responding more slowly and selective to some specific spectro-temporal features, indicating a narrower receptive field, which we model as gamma excitatory neurons.

The authors did not report a spatial dissociation between the two subclasses, leaving the possibility open that theta and gamma circuits do indeed coexist in one place. This point has been added to the Result section of the manuscript:

"It also mirrored the dissociation in primate auditory cortex between a population of 'stereotyped' neurons responding very rapidly and non-selectively to any acoustic stimulus (putatively *Te* neurons) and a population of 'modulated' neurons responding selectively to specific spectro-temporal features (putatively *Ge* neurons) (8)."

*4B) A major issue raised consistently in the reviews is that throughout the paper, the specific statistical tests applied were opaque or missing. Examples include p values (or Bayes factors, etc.) to back up many of the comparisons across conditions, syllable boundary detection (is the presented model's performance significantly better than the others?), and decoding of simple stimuli (again, is performance significantly better for the intact model?). What is the chance level of decoding for the conditions shown in*
Figure 3
*and*
Figure 4*? These are not necessarily what might be expected, depending on the number of samples available. Although the approaches seem generally reasonable, statistical support must be reported (incorporating, if applicable, correction for multiple comparisons)*.

We thank the reviewers for this comment, which urged us to provide more statistical details. We have now included results from statistical testing at all steps of the analyses. Statistical effects were very strong in all cases and reported both below and in the main manuscript. For the case of chance levels, given that we use 10 classes for all classification analysis, the average expected chance level would be 10%. Based on Combrisson and Jerbi, the threshold for 5% testing would be resp. 12.2% and 11.6% for each sawtooth (500 samples) and syllable (1000 samples) classification analysis. Importantly here, the classification procedure was repeated many times (10 times for sawtooth, 200 times for syllables) with a different set of samples. Hence the distribution of classification score could be compared through a simple t-test against its expected mean value for the chance level (10%).

Details of the statistical results are as follows:

Spike phase-amplitude coupling: coupling was significant for the full model both for rest and speech (p< 10^-9^ for both);

Syllable boundary detection performance (in the subsection headed “Syllable boundary detection by theta oscillations”): the theta network performed significantly better than the Mermelstein and LN algorithms and than chance level for uncompressed and compressed speech (all p-values < 10^-12^, tested over 3620 sentences);

Sawtooth decoding (in the subsection headed “Decoding of simple temporal stimuli from output spike patterns”): decoding using the full model was significantly larger than using any of the two control networks either using stimulus timing or theta timing (all p-values <10^-9^, testing over 10 repetitions of the classification procedure);

Syllable decoding (in the subsection headed “Continuous speech encoding by model output spike patterns”): in the full model, decoding was lower using spike count than using spike patterns (p<10^-12^ over 200 repetitions); decoding using any of the control models and any of the neural code (spike count/spike patterns) was significantly lower than using spike patterns for the full model (all p-values <10^-12^), and none was significantly better than using spike counts for the full model (all p-values>.08 uncorrected);

Classifier analysis (in the subsection headed “Classifier analysis”): decoding using spike counts was significantly smaller than using spike patterns (p<10^-12^ over 200 repetitions for all 6 window sizes); the increase in decoding performance using phase-partitioned code rather than spike count was significantly reduced in both control models and all 6 window sizes compared to full model (all p-values<10^-12^);

Mutual Information (in the subsection headed “Mutual information”): in the full model, MI was significantly larger when adding phase-partitioned code on top of both spike count and time-partitioned code (p<10^-12^ with vs without phase information (p-values over 32 *Ge* neurons: p<10^-12^ for both).

*5) Some additional comment on phase-reset is warranted. For example,*
Figure 2
*nicely shows PINTH activity (LFP and theta-associated activity) occurring in a (quasi-)regular fashion before the presentation of the speech stimulus. The degree to which the incoming speech signal systematically alters the phase of this activity is unclear: from*
Figure 2*, it appears to happen instantaneously, with no missed/extra spikes. Such performance seems intriguing but potentially unrealistic. How does the phase following speech presentation compare with that prior to speech presentation*?

The strong speech input to *Te* neurons provides a somewhat hard reset of theta phase in correspondence of a sentence onset. Unlike the weak coupling situation, the theta phase after the onset is virtually independent of the phase prior to speech onset. This mechanism enables very rapid and strong phaselocking of the theta to speech throughout the sentence. Gross et al. (Plos Biology, 2013 showed that phaselocking to speech edges occurs in auditory cortex within a few hundred milliseconds. To better illustrate this important property, we added a figure (Figure 1—figure supplement 1) that shows theta phase concentration for multiple presentation of the same sentence: there is a rapid transition within a few hundred ms (e.g. a theta cycle or less) from uniform phase distribution before sentence onset to very strong phase-locking. The figure is referenced in the Results section “Model dynamics in response to natural sentences”.

*6) Please check figures and figure captions. For example, there are captions for*
Figure 2
*C, D and E, but these panels are missing in*
Figure 2*. (Note also a similar mismatch for*
Figure 2—figure supplement 1.*)*

This was an error. This has been corrected for Figure 2 and Figure 2—figure supplement 1. The legends for two panels of Figure 5—figure supplement 1 have also been correctly reordered.

[Editors' note: further revisions were requested prior to acceptance, as described below.]

*1) Please address the following in the paper itself. The comment is in response to a point made in the authors' cover letter*.

*A system without noise would always perform at 100% on any classification test if given repeats of certain items in training, and then tested on those same items. Your point that noise is needed to optimize detection of stimulus onset time is important and should be included in the paper (that is, for tests in which stimulus onset time is not provided to the model)*.

We agree with reviewers that this point should be made explicit in the manuscript. We have now added in the Results section:

"Noise in the theta module allows the alignment of theta bursts to stimulus onset and thus improves detection performance by enabling consistent theta chunking of spike patterns."

*2) Given the authors' intent to take into account the constraint of neural noise, additional information as to how well this is done should be provided. At a minimum, in the spiking models, please provide the CV of the spike trains and the Fano factors for responses to identical stimuli. For the LFP signals, also provide an indicator of the trial-to-trial variability in response to identical stimuli: e.g., when stimuli are identical, CV of heights of particular LFP peaks, or of time intervals between specific peaks and troughs*.

We have added (in Figure 1—figure supplement 1) spike train CV and spike count Fano factors for the different types of neurons in response to speech as well as the standard deviation of LFP in response to speech. The latter shows a great decrease in LFP variability at sentence onset that is mostly due to the phase-locking of theta and gamma oscillations. We added reference to both new panels in the Results section.